# Molecular-Biology-Driven Frontline Treatment for Chronic Lymphocytic Leukemia: A Network Meta-Analysis of Randomized Clinical Trials

**DOI:** 10.3390/ijms24129930

**Published:** 2023-06-09

**Authors:** Andrea Rizzuto, Angelo Pirrera, Emilia Gigliotta, Salvatrice Mancuso, Candida Vullo, Giulia Maria Camarda, Cristina Rotolo, Arianna Roppolo, Corinne Spoto, Massimo Gentile, Cirino Botta, Sergio Siragusa

**Affiliations:** 1Department of Health Promotion, Mother and Child Care, Internal Medicine and Medical Specialties (ProMISE), University of Palermo, 90141 Palermo, Italy; 2Hematology Unit, “Annunziata” Hospital of Cosenza, 87100 Cosenza, Italy; 3Department of Pharmacy, Health and Nutritional Sciences, University of Calabria, 87036 Rende, Italy

**Keywords:** CLL, chronic lymphocitic leukemia, network metanalysis, Bruton’s tyrosine kinase inhibitors

## Abstract

The treatment of chronic lymphocytic leukemia (CLL) currently relies on the use of chemo-immunotherapy, Bruton’s tyrosine kinase inhibitors, or BCL2 inhibitors alone or combined with an anti-CD20 monoclonal antibody. However, the availability of multiple choices for the first-line setting and a lack of direct head-to-head comparisons pose a challenge for treatment selection. To overcome these limitations, we performed a systematic review and a network meta-analysis on published randomized clinical trials performed in the first-line treatment setting of CLL. For each study, we retrieved data on progression-free survival (according to del17/P53 and IGHV status), overall response rate, complete response, and incidence of most frequent grade 3–4 adverse event. We identified nine clinical trials encompassing 11 different treatments, with a total of 5288 CLL patients evaluated. We systematically performed separated network meta-analyses (NMA) to evaluate the efficacy/safety of each regimen in the conditions previously described to obtain the surface under the cumulative ranking curve (SUCRA) score, which was subsequently used to build separated ranking charts. Interestingly, the combination of obinutuzumab with acalabrutinib reached the top of the chart in each sub-analysis performed, with the exception of the del17/P53mut setting, where it was almost on par with the aCD20 mAbs/ibrutinib combination (SUCRA aCD20-ibrutinib and O-acala: 93.5% and 91%, respectively) and of the safety evaluation, where monotherapies (acalabrutinib in particular) gave better results. Finally, considering that NMA and SUCRA work for single endpoints only, we performed a principal component analysis to recapitulate in a cartesian plane the SUCRA profiles of each schedule according to the results obtained in each sub-analysis, confirming again the superiority of aCD20/BTKi or BCL2i combinations in a first-line setting. Overall, here we demonstrated that: (1) a chemotherapy-free regimen, such as the combination of aCD20 with a BTKi or BCL2i, should be the preferred treatment choice despite biological/molecular characteristics (preferred regimen O-acala); (2) there is less and less room for chemotherapy in the first line treatment of CLL.

## 1. Introduction

Chronic lymphocytic leukemia (CLL) is a lymphoproliferative disorder that accounts for about 30% of adult leukemias and 25% of non-Hodgkin’s lymphomas (NHL) [1]. In the last update of the SEER database, the age-adjusted incidence of CLL was 4.9 per 100,000 inhabitants per year [2], making it one of the most common types of leukemia. From a biological point of view, CLL is often characterized by the loss or addition of large chromosomal material (e.g., del13q, del 11q, and trisomy 12), as well as specific point mutations which increase the aggressivity of the disease [3]. Approximately 80% of all patients with CLL have at least one of four common chromosomal alterations: a deletion in chromosome 13q14.3 (del(13q), del(11q), del(17p)), or trisomy 12 [4]. The evaluation of chromosome 17p deletion and/or TP53 mutation, as well as IGHV hyper-mutational status, is currently recommended in each patient before starting a new treatment for CLL due to their significant impact on treatment response [5].

Loss of p53 activity can be due to either somatic alterations in the gene or chromosomal alterations involving it, even in association. These alterations are less frequent in newly diagnosed patients than in those relapsed or with progression/transformation to aggressive lymphoma. However, the presence of the del17/TP53mutation, even in newly diagnosed patients, is indicative of poor outcome, with a median survival between 3 and 5 years, and worse response to first-line chemo-immunotherapies [5]. Interestingly, it has been demonstrated that the progression-free survival and overall survival of CLL patients carrying a del(17p) and patients carrying a TP53 mutation are similar [6]. Additionally, unmutated IgHV status, relating to shorter lymphocyte doubling time, CD38 overexpression, and adverse karyotype [7], is associated with a more aggressive course of CLL compared to patients with mutated IgHV, with a PFS in the range of 1 to 5 years compared to a range of 9.2 to 18.9, and a median OS of 3.2–10 years vs. 17.9–25.8 years, respectively [8].

According to the iwCLL guidelines, in CLL patients, a first-line treatment should be promptly initiated once specific clinical or laboratory events happen: including evidence of progressive bone marrow failure, progressive or symptomatic splenomegaly or lymphadenopathy, an increase in lymphocyte count ≥50% in a 2-month period, or systemic symptoms. Additionally, these patients, once the treatment start has been planned, should be evaluated (at least) for the presence of the del17/TP53 mutation and hyper-mutational status of IGHV in order to identify the most suitable treatment option among chemo-immunotherapy or chemo-free regimens (BTK, BCL2 or PI3K inhibitors), even if a specific guideline still lacks [9].

As new agents have emerged for the treatment of CLL, the optimal therapeutic combination strategies according to the current molecular stratification have yet to be established [10]. On these bases, we performed a network meta-analysis to integrate all the clinical evidence in the first-line treatment of CLL with the aims of identifying the best regimen(s) for each specific molecular subgroup in terms of efficacy and safety.

## 2. Results

### 2.1. Trials Characteristics

A total of 9 studies were included within the meta-analysis, reporting 11 different treatment schedules: ibrutinib, acalabrutinib, a combination of an anti-CD20 (rituximab or Obinutuzumab) and ibrutinib (antiCD20-ibrutinib), obinotuzumab-acalabrutinib, obinutuzumab-venetoclax, obinutuzumab-chlorambucil, chlorambucil, rituximab-chlorambucil, rituximab-bendamustine, fludarabine-cyclophophamide (FC), and rituximab-fludarabine-cyclophosfamide (FCR) [11,12,13,14,15,16,17,18,19] (Table 1, Figure 1A). The total number of patients evaluated in this NMA was 5288. All studies evaluated satisfied the inclusion criteria previously described. Missing information in early trial reports were calculated or obtained from subsequent updates. No significant inconsistencies or loop-specific heterogeneities were found in our NMA (Appendix A). For the purpose of statistical calculation, treatments including the combination of rituximab or obinutuzumab and ibrutinib were considered in aggregate as a combination of anti-CD20 antibody and ibrutinib, based on the results of second-line and follow-up studies [20].

### 2.2. PFS Analysis

#### 2.2.1. Del17/p53mut Population

The analysis regarding Del17/p53mut patients included five different trials for a total of nine treatments (Figure 1). Within some studies, high-risk populations that also presented other mutations described as such in the literature were taken into account, since HR for PFS of the individual populations examined were not available or obtainable. The calculation algorithm identified three triangular loops (Figure 1B and ). We used the chlorambucil arm as a comparator for all other schedules. The highest SUCRAs were reported by anti-CD20 + ibrutinib and obinutuzumab-acalabrutinib treatments (SUCRA 93% and 91%, respectively), followed by both BTKi monotherapies (ibrutinib and acalabrutinib, 78% and 62.5%, respectively). Nevertheless, the superiority of the anti-CD20-BTKi combination is overwhelming. The worst treatments were chlorambucil as monotherapy (SUCRA 0%) and the rituximab + chlorambucil association (SUCRA 12.5%). Finally, the association of Obinutuzumab + Venetoclax reported a SUCRA of 42.2% (Figure 1B and Appendix A).

#### 2.2.2. No Del17/p53wt Population

The analysis regarding patients Del17/p53wt included a total of 11 treatments. The algorithm identified three triangular loops and one quadratic loop (Appendix A). Once again, the best treatment that emerged from the analysis was the combination of anti-CD20 + BTKi (obinutuzimab + acalabrutinib SUCRA 96.8%, aCD20 + ibrutinib 68.3%), followed by BTKi monotherapy (acalabrutinib SUCRA 87%, Ibrutinib SUCRA 52.8%). The other target therapy analyzed, the obinutizimab + venetoclax combination, was effective in this patient population, with a SUCRA of 81.6% (vs. SUCRA 42.2% in the17/p53mut patients). Once again, the worst treatment was chlorambucil monotherapy (SUCRA 3.6%), closely followed by the other chemotherapy regimens analyzed (both in monotherapy and associated with immunotherapy, with a superiority of the latter) (Figure 1B and Appendix A).

#### 2.2.3. IgHV-Hypermutated Population

The analysis regarding IgHV-hypermutated patients encompassed a total of 11 treatments. The algorithm identified four triangular loops and one quadratic loop (Appendix A). The superiority of aCD20-BTKi combination treatments is still a constant, with the SUCRA of aCD20 + ibrutinib and obinutuzumab + acalabrutinib at 79.8% and 99.2%, respectively. Interestingly, in this particular group, we observed an increased SUCRA for chemoimmunotherapy combinations compared to the other molecular groups analyzed (SUCRA for rituximab-bendamustine: 53%, FCR: 35.5%). Additionally, in this molecular population, the obinutuzumab + venetoclax combination achieved a greater SUCRA than BTKi in monotherapy (86.4%, different from what was seen in the del17/p53mut and p53wt populations). Again, the less effective treatments were chemotherapies (chlorambucil SUCRA 0.4, FC SUCRA 14%) (Figure 1B and Appendix A).

#### 2.2.4. IgHV-Germline Population

In the last subpopulation analyzed for PFS, 11 treatments were evaluated. The algorithm identified four triangular loops and one quadratic loop (Appendix A). This analysis confirms the higher efficacy of new target drugs as compared to chemotherapy: obinutuzumab + acalabrutinib reported a SUCRA of 97.5%, while the best chemo/chemoimmunotherapy combination, FCR, had a SUCRA of 34.6% (Figure 1B and Appendix A).

### 2.3. ORR and CRR

The analysis on ORR and CRR were carried out without considering the risk subgroups due to the lack of specific data for each molecular subgroup. Eleven treatments were evaluated. Interestingly, while target treatments reported constant higher SUCRAs considering ORR (obinotuzumab-acalabrutinib combo reported an impressive SUCRA of 96%), chemoimmunotherapy treatments reported the highest SUCRAs in term of CRR (FCR SUCRA: 83.9%), followed by target treatments (Figure 2 and Appendix A).

### 2.4. Safety

Lastly, we performed an NMA analysis to evaluate the safety of all treatments. The most frequent G3/G4 adverse event reported in the studies was neutropenia. The safest treatment proved to be acalabrutinib as monotherapy (SUCRA 99.6%), followed by ibrutinib monotherapy (SUCRA 85.2%). The addition of an AntiCD20 to BTKi demonstrated, as expected, higher rates of AEs than their monotherapy counterparts. Chlorambucil as monotherapy was shown to be safer than the rest of the chemo- and chemoimmunotherapy treatments (chlorambucil SUCRA 78%, rituximab-bendamustine SUCRA 61.8%, obinutuzumab + chlorambucil SUCRA 22.9%, FCR SUCRA 13.5%, FC SUCRA 0.3%). In contrast, the obinutuzumab + venetoclax combination reported a SUCRA of 36.3%, lower than the rest of the target therapies both in monotherapy and in combination treatment (Figure 1B and Appendix A).

### 2.5. Results Integration by Principal Component Analysis

As a final analysis, we performed an integration of all the obtained results. Firstly, we performed a ranking according to the mean SUCRA obtained for each variable analyzed (Figure 3). As expected, the combination of O-Acalabrutinib reported the overall highest benefit in terms of efficacy and safety. However, the mean value does not perfectly recapitulate the whole activity profile of a treatment schedule. Along this line, we performed a principal component analysis (PCA) by evaluating all treatments under study and distributing them within a two-dimensional plane according to their SUCRA profile. The distance between individual points reflects the distance between the SUCRA profiles of each schedule. In order to include all treatments evaluated in our study, we performed two different PCA: one excluding del17/p53mut patients and a second including all molecular subgroups (which excluded FC and FCR treatments, not evaluated in del17/p53mut patients). Subsequently, we obtained an unsupervised clustering of all the treatments under investigation into two final groups. In the PCA including all molecular subgroups, all target treatments, both as monotherapy and associated with antiCD20, were found to be very close, reflecting their high efficacy and safety compared to chemoimmunotherapy treatments (Figure 4). The difference between these two clusters becomes even greater when all treatments are evaluated together. This second PCA shows that target treatments (such as BTKi with or without antiCD20 and obinutuzumab + venetoclax) are more likely to show benefit as first-line treatments than chemo- and chemoimmunotherapy treatments (FCR, FC, chlorambucil, antiCD20 + chlorambucil, rituximab + bendamustine) (Figure 4). Of interest, even in this case, O-Acalabrutinib performed better than other treatments, showing the best efficacy coupled with an acceptable safety profile.

## 3. Discussion

The advent of BTKi and BCL2 inhibitors for the treatment of CLL have opened the way to new therapeutic avenues and alternative drug combinations to chemotherapy, with the aim of improving safety while ensuring better long-term disease control. With our study, we confirm the advantage of using target therapies in the frontline setting; specifically, ibrutinib, acalabrutinib, and venetoclax in combination with rituximab and obinutuzumab reached the top scores in term of outcome, despite molecular profiling. Additionally, chemoimmunotherapy reached comparable results in terms of complete response rates only. These results are in line with recent studies in this field [22,23]. Unfortunately, due to the differences in follow-up times, most of the results related to innovative schedules should be proved and confirmed over the coming years. Indeed, the top-ranking schedule, Obinutuzumab-acalabrutinib (ELEVATE-TN) [14], while promising a median PFS exceeding 100 months, still lacks a confirmed median PFS. A further interesting point emerging from our results is a challenge to the current indication related to the need for appropriate molecular characterization (del17/p53 mutation or IGHV mutation study) before starting a frontline regimen. Indeed, even in “high risk” patients, the use of BTKi alone or in combination with anti-CD20 performed way better than chemo/immunochemotherapy, moving the molecular characterization from a “predictive” to an exclusively “prognostic” role. This point applies to the small subgroup of IGHV hypermutated fit patients, where if confirmed, the role of new target therapy seems comparable (if not superior) to chemotherapy regimens such as FC/FCR [19]. On the other hand, while challenging molecular definition, our results point attention to the long-term sustainability of these treatments. Indeed, the identified “one-size-fit-all” treatment strategies carry an important (even economic) draw-back related to the need for a continuous use of the drugs (until disease progression or unacceptable toxicity), whereas chemotherapy and immunotherapy (even if combined with venetoclax) are used for fixed periods [11,12,13,14,15,16,17,18,19]. Additionally, it should be taken into account that several differences in drug use between countries exist; i.e., it is possible in the USA (but not in Europe) to re-treat patients on venetoclax after time-limited therapy, thus potentially increasing the benefits of first-line use of this drug. Thus, cost-utility analyses are warranted in this setting to identify the best schedule. Surprisingly, within the BTKi group, SUCRA analyses demonstrated substantial equivalence in efficacy between ibrutinib and acalabrutinib, with a significant improvement in responses when an anti-CD20 antibody is combined with both. The slight differences in SUCRA within groups (monotherapy and combinations) are likely to be dependent on fluctuations in the data and the number of patients involved in the analysis. We further observed an important difference in efficacy between BTKi and venetoclax in the del17/p53mut subgroup. Interestingly, in these patients’ group, venetoclax performs similarly to chemotherapy; it is therefore likely that the absence of a functional p53 translates into a critical lack of an important regulatory pathway of the apoptotic process, thus limiting the significance of BCL2 inhibition, as observed in studying interactions between ABT-737-induced apoptosis and chromosome 17 deletion in CLL cells [24]. On the other hand, the regulatory activity of intracellular signalling promoted by Bruton’s tyrosine kinase is barely affected by this phenomenon. The impact of these findings on the development of molecularly driven sequence strategies warrant specific investigation in translational clinical trials. Nevertheless, these data support the idea that if del17/p53 mutational status is not available, a BTKi with or without an anti-CD20 mAbs represents the best frontline treatment choice. Regarding the choice between rituximab and obinutuzumab, and specifically, which of the two provides the best efficacy results, is still matter of debate. In this meta-analysis, we considered ibrutinib/anti-CD20 combinations to be equivalent, although some preliminary and not fully validated data support the idea that obinutuzumab is generally more effective than rituximab; in the CLL11 study, where the two antibodies were combined with chemotherapy, obinutuzumab was superior to rituximab in all subgroups except in del17 patients [21].

To reduce the bias derived from using these analytical techniques to identify the overall best treatment, we used PCA methods to generate subgroups including treatments with homogeneous results; this approach offers an armamentarium of alternative schedules that can be used according to physician and patient needs. The unsupervised hierarchical clustering applied to PCA led to the identification of two well-separated groups composed almost exclusively by target therapies or chemotherapies. The latter included the worse SUCRA scores overall, supporting the idea that, despite patients’ molecular signature, there is even less room for chemotherapies in the treatment strategy of CLL patients.

However, our analysis does have certain limitations. Firstly, all data used in our study were derived from published clinical trials rather than from individual patient data. Moreover, there is significant heterogeneity in the populations across different studies, which may potentially affect the generalizability of our findings. Additionally, some treatments included in the study were based on a single trial, which could limit the robustness and reliability of our findings. Having multiple studies for each treatment would have improved the accuracy and confidence of our analysis.

Despite this, the data reported here could be of help in optimizing treatment choices algorithms for patients with treatment-naive CLL. Response rates were analyzed as aggregated data: the absence of specific data related to molecular subgroups made it impossible to proceed with a deep analysis (as performed with PFS). Secondly, data on overall survival were not considered in this analysis, mainly due to poorly represented and immature data. As a result, we are currently unable to determine whether target therapy, while apparently better in achieving disease control, is actually effective in increasing patient OS over the long term, or whether—despite being tolerated well—long-term treatment affects patients’ survival. On this term, the recent update on the E1912 study demonstrates that the association treatment of BTKi and anti-CD20 antibodies (ibrutinib and rituximab in this specific case) performed better than chemoimmunotherapy, even in terms of OS, limiting even more the role of chemotherapy for these patients [25].

Overall, our study demonstrated that the use of target therapies is probably preferable to the use of chemotherapy in the first-line treatment of CLL despite molecular characterization, having demonstrated greater efficacy and acceptable safety. While a small clinical significance in specific groups of patients not eligible for BTKi or venetoclax, or requiring rapid disease control, could be found, and while the presence of a del17/p53 mutation could identify venetoclax-poor-responder patients, no clear role for baseline molecular characterization or for the use of chemotherapy regimens could be supported on the bases of our results. Further studies on the identification of new molecular predictive factors as well as new targets are eagerly awaited.

## 4. Materials and Methods

### 4.1. Search Strategy

The PubMed database was interrogated for the identification of studies to be included into the statistical analysis by including the following research keys in different combinations: “chronic lymphocytic leukemia”, “CLL”, “first-line treatment”, “treatment-naïve”, “untreated”. The identified articles were subsequently screened, and their abstracts were read to investigate if inclusion criteria were met. The last date of the search was 31 December 2021 (Figure 5).

### 4.2. Inclusion and Exclusion Criteria

Identified trials were included in the analysis if: (1) they were comparative phase 2/3 studies; (2) they included patients with chronic lymphocytic leukemia who had never received treatment; and (3) data regarding the required endpoints (progression free survival (PFS), complete response (CR), overall response rate (ORR) and risk ratio (RR) of most frequent treatment-related G3-4 toxicity) were included or if it was possible to derive them from published data. Studies that considered patients beyond first-line treatment, included patients on both first- and second-line treatment, and cross-trial comparisons were not included.

### 4.3. Data Extraction

For each study, we evaluated: (1) hazard ratio (HR) of PFS; (2) odds ratio (OR) of ORR and CR; (3) RR for safety (assessment of the most common grade 3–4 toxicity); Data regarding HR for PFS were evaluated in relation to 4 main patient populations: patients carrying chromosome 17 (del17) deletion and/or p53 mutation (p53mut), patients not carrying del17/p53mut, patients carrying immunoglobulin heavy chain (IgHV) gene hypermutation, and patients with germline status of IgHV. If the HR of survival curves was not reported, it was derived from the graph by using the method of Tierney et al. [26]. Studies that did not report PFS data for any of the above categories were not included within the analysis for those specific categories. The title, first author, publication date of the study, number of patients examined, and number of patients included within the 4 categories under analysis were also extrapolated from these studies. Data regarding ORR, CR, and AE were analyzed on the whole intention to treat the population of each study.

### 4.4. Network Meta-Analysis

For the analytical part, we used Bayesian analysis to compare multiple treatment regimens as described elsewhere [27,28,29]. Briefly, the analysis was performed in STATA software using the “mvmeta” package. Specifically, the NMA synthesizes data from a network of studies involving multiple interventions, and thus, by integrating direct and indirect comparisons, has the potential to rank treatments by outcome. Here, we ranked the evaluated regimens according to PFS, ORR, CR, and incidence of grade 3–4 adverse events. The relative effects of the treatments were reported with hazard ratios for PFS and odds ratios and risk ratios for ORR, CR, and AE, along with their 95% confidence interval. From these elements was calculated the surface under the cumulative ranking curve (SUCRA), the most important value useful to classify the evaluated treatments; the closer the value of SUCRA is to 1, the higher the probability that the treatment is the best compared to others. Concurrently, we assessed the heterogeneity and consistency of our analysis by evaluating the log of the ratio of 2 odds ratios (RoR) from direct and indirect evidence in the loop (ifplot command in STATA) [30,31]. We performed an NMA with an (RE) model by using a Markov chain Monte Carlo simulation technique with up to 10,000 iterations for each prespecified outcome. At the end of the analysis, all treatments had 6 to 7 evaluable SUCRAs. To reduce the bias associated with single analyses, we conducted a principal component analysis in R (prcomp) by considering all SUCRAs from individual treatments and distributing them within a plan [29]. This allowed us to group treatments based on similarities in efficacy and safety, and therefore allowed us to evaluate multiple high-benefit treatments rather than one treatment that was better than all others [32,33].

## Figures and Tables

**Figure 1 ijms-24-09930-f001:**
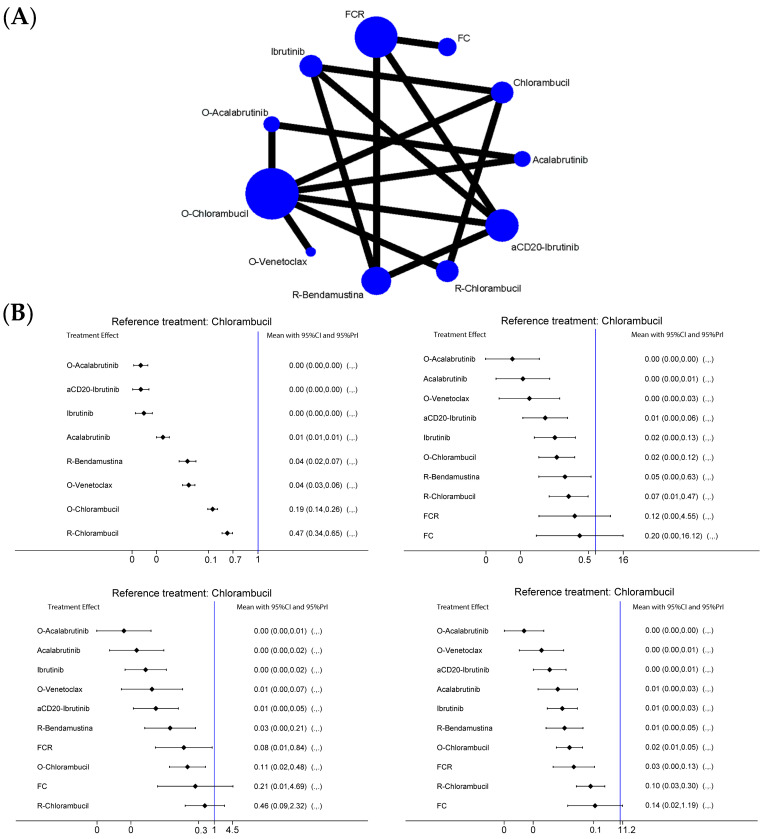
Network plot displaying all treatments included in the analysis for all endpoints. (**A**) Effect estimates of the treatments in terms of progression-free survival (PFS) based on risk classes (ordered from top left, clockwise: p53 mutated population, p53 unmutated population, germline IgHV population, and hypermutated IgHV population) using chlorambucil as a comparator: the less the effect, the more it is in favor of the comparator against the reference. (**B**) O: Obinutuzumab; R: Rituximab; aCD20: Obinutuzumab/Rituximab; FCR: fludarabine-cyclophosphamide-rituximab; FC: fludarabine-cyclophosphamide.

**Figure 2 ijms-24-09930-f002:**
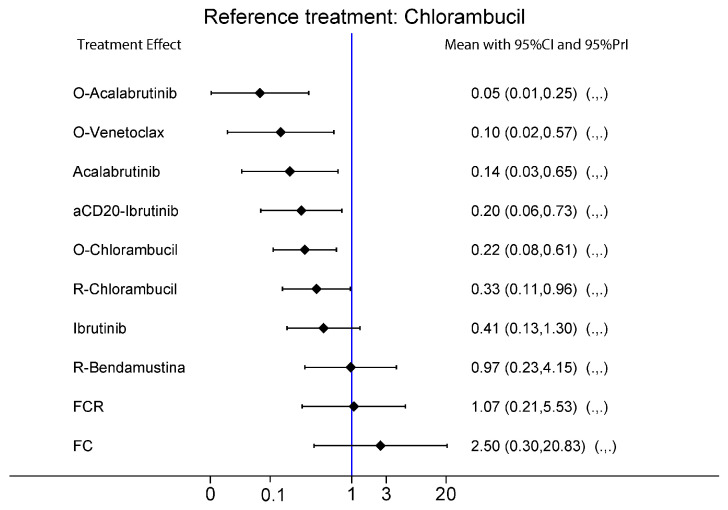
Effect estimates of the treatment in terms of overall response rate, overall complete response, and safety (Top–down: ORR, CR, safety) on all patients using chlorambucil as comparator. O: Obinutuzumab; R: Rituximab; aCD20: Obinutuzumab/Rituximab; FCR: fludarabine-cyclophosphamide-rituximab; FC: fludarabine-cyclophosphamide.

**Figure 3 ijms-24-09930-f003:**
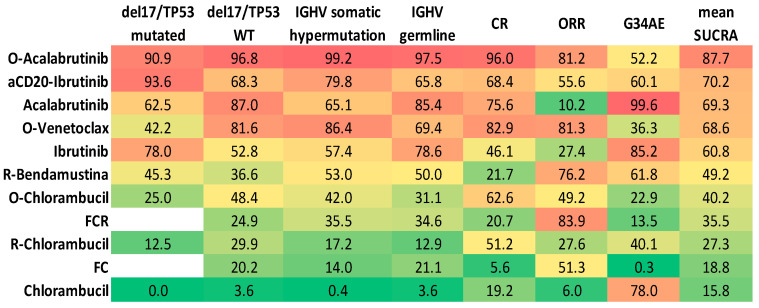
Heatmap reporting the SUCRA scores for each endpoint, including a mean of all SUCRAs in the last column. The green color represents the lowest SUCRA. Regimens are ordered according to mean SUCRA from the highest to lowest.

**Figure 4 ijms-24-09930-f004:**
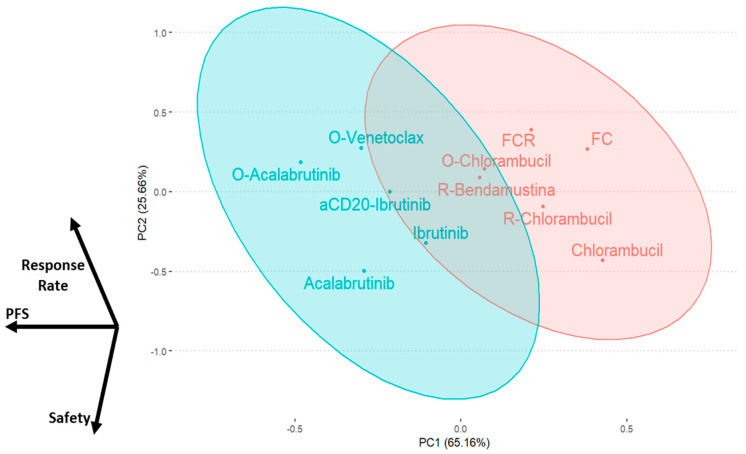
Principal component analysis reporting all the regimens analyzed, grouped (unsupervised clustering) according to their SUCRA profile.

**Figure 5 ijms-24-09930-f005:**
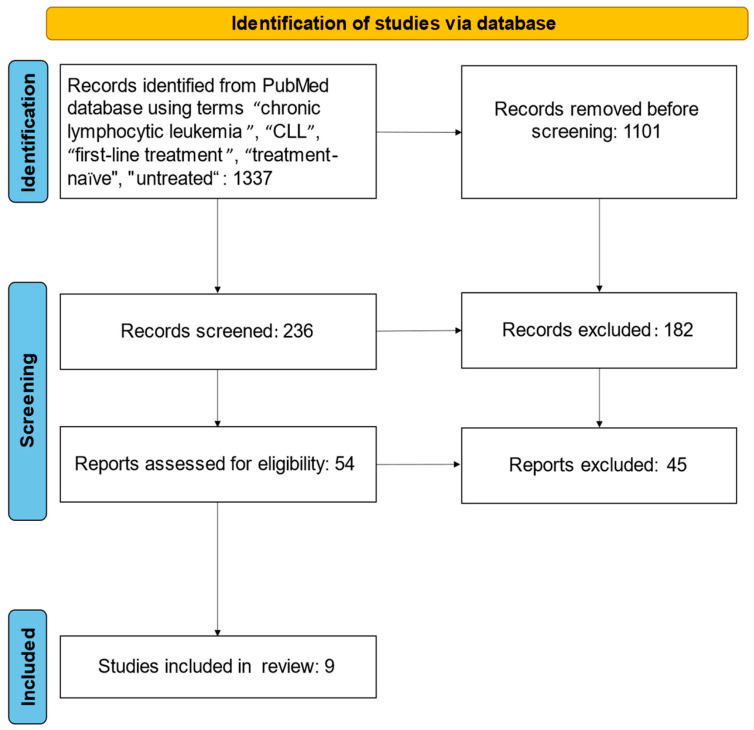
PRISMA flow-chart.

**Table 1 ijms-24-09930-t001:** List of all trials included in the network meta-analysis (NMA).

Trials	Authors	Treatments
Resonate II	Burger et al. [13]	Ibrutinib vs. Chlorambucil
A041202	Woyach et al. [11]	Rituximab-Ibrutinib vs. Ibrutinib vs. Rituximab-Bendamustine
E1912	Shanafelt et al. [12]	Rituximab-Ibrutinib vs. Fludarabine-Cyclophosphamide-Rituximab
iLLUMINATE	Moreno et al. [18]	Obinutuzumab-Ibrutinib vs. Obinutuzumab-Chlorambucil
ELEVATE-TN	Sharman et al. [14]	Obinutuzumab-Acalabrutinib vs. Acalabrutinib vs. Obinutuzumab-Chlorambucil
CLL-14	Al-Sawaf et al. [15]	Obinutuzumab-Venetoclax vs. Obinutuzumab-Chlorambucil
CLL-10	Eichorst et al. [16]	Fludarabine-Cyclophosphamide-Rituximab vs. Rituximab-Bendamustine
CLL-11	Goede et al. [21]	Obinutuzumab-Chlorambucil vs. Rituximab-Chlorambucil vs. Chlorambucil
CLL-8	Hallek et al. [19]	Fludarabine-Cyclophosphamide-Rituximab vs. Cyclophosphamide-Rituximab

## Data Availability

Not applicable.

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
