# Peer review of "Molecular-Biology-Driven Frontline Treatment for Chronic Lymphocytic Leukemia: A Network Meta-Analysis of Randomized Clinical Trials"

_ijms, 2023, doi:10.3390/ijms24129930_

Round 1
Reviewer 1 Report
In this manuscript by Rizzuto et al, the authors attempt to answer a crucial question in the CLL field of the individual impact of cytogenetic biomarkers on the outcomes of individual therapies by taking a data meta-analysis approach to large front line studies of CLL. The manuscript is generally well written with with good presentation of the data. I have the following minor comments:
In the title: Molecular biology driven frontline treatment for chronic lymphocytic leukemia: the end of an era? ---it is unclear what “the end of an era” refers to from the manuscript—please remove “the end of an era”
Please expand NMA in the abstract
Please remove the u from “remouved”
Please indicate as one of the limitations that the analysis for some of the treatments only rely on one study, for example, in the case of venetoclax + obin as well as acalabrutinib, so incorporating more studies will be necessary to increase the robustness of the results.
Author Response
Q: In this manuscript by Rizzuto et al, the authors attempt to answer a crucial question in the CLL field of the individual impact of cytogenetic biomarkers on the outcomes of individual therapies by taking a data meta-analysis approach to large front line studies of CLL. The manuscript is generally well written with with good presentation of the data. I have the following minor comments:
A: We thank the reviewer for these positive comments, we hope the revised version of the manuscript, including your suggestions, will met your expectations
Q: In the title: Molecular biology driven frontline treatment for chronic lymphocytic leukemia: the end of an era? ---it is unclear what “the end of an era” refers to from the manuscript—please remove “the end of an era”
A: We thank the reviewer for this suggestion, and ,accordingly, we changed the tilte in “Molecular-biology-driven frontline treatment for chronic lymphocytic leukemia: a network meta-analysis of randomized clinical trials” to improve the overall clarity.
Q: Please expand NMA in the abstract
A: Done as requested
Q: Please remove the u from “remouved”
A: Done as requested
Q: Please indicate as one of the limitations that the analysis for some of the treatments only rely on one study, for example, in the case of venetoclax + obin as well as acalabrutinib, so incorporating more studies will be necessary to increase the robustness of the results.
A: We thank the reviewer for this important consideration. We included this limitation within our discussion.
Reviewer 2 Report
Interesting meta-analysis regarding new therapeutic strategies in the chronic lymphatic leukemia landscape. The work is precise, well written and organized.
A few minor observations, especially on the figures and their captions
- I think Figure 1 needs to be revised overall
- Lines 183-186, please specify SUCRA of antiCD 20-venetoclax population
- Figure 2, I suggest improving the figure caption
- Figure 4, I suggest explaining better the meaning of the figure
Thank you and congratulations!
Author Response
Q: Interesting meta-analysis regarding new therapeutic strategies in the chronic lymphatic leukemia landscape. The work is precise, well written and organized.
A: We sincerely thank the reviewer for these positive comments.
Q: A few minor observations, especially on the figures and their captions
- I think Figure 1 needs to be revised overall
A: We apologize for the lack of clarity and overall poor quality of the Figure 1. We included now an updated Figure 1 in the revised version of the manuscript
Q Lines 183-186, please specify SUCRA of antiCD 20-venetoclax population
A: Done as requested
Q Figure 2, I suggest improving the figure caption
A: We apologize for the lack of clarity. Done as requested.
Q Figure 4, I suggest explaining better the meaning of the figure
A: We apologize for the lack of clarity. Done as requested.
Reviewer 3 Report
The article provided sound information and well illustrated
Author Response
Q: The article provided sound information and well illustrated
A: We really appreciate the endorsement received from the reviewer and we thank him for the support to our work.
Reviewer 4 Report
In the era of modern societies, stress is a part of our life process. As per recent discoveries, Stress is the most important factor, which could induce tumorigenesis and promote cancer development via creating mutations and modulation of genetic & epigenetic activities. Stress has the potential power to generate almost all the hallmarks of cancer in a healthy person. Moreover, it could modulate the chemotherapeutic efficacy via the regulation of differential gene expressions in different types of cancer (https://doi.org/10.3389/fonc.2020.01492; https://doi.org/10.1038/s41568-021-00395-5).
Not only cancer, stress could create all kind of human disease depending on the level of stress and duration of stress. So stress is an important source of creation of all kind of problem in this sophisticated human system.
In line with above discoveries, few other studies are showing restoration of gene expression and better therapeutic efficacy after reducing stress in the system by applying different stress management strategies. (https://doi.org/10.1073/pnas.2110455118 ; DOI 10.1007/s11764-012-0252-8, https://doi.org/10.1016/j.urolonc.2020.09.011 ; https://doi.org/10.1177/15347354209099 )
Considering the above fact, it is logical that if the source of creation,’Stress’ could be reduced in the system, any bad drug could work well. On the contrary, if stress level increases in the system, any good drug will show lower efficacy. Even, disease may not manifest if the stress could manage properly. This might be a main cause to show differential efficacy of one drug in different country depending on the country specific stress level.
a) So please re analyze and stratify your data considering past or present stressful life event as one of the factor.
b) If stress information is not available, please collect new set of data from hospital record and reanalyze.
c) Your analysis is mainly based on published database. Please validate your findings by collecting new patients information from your regional hospital.
d) In discussion section, please modify your discussion considering stress and its management related to different treatment regiment.
Author Response
According to Editor indication, this page has been left blank.